# Time Dependency of Current Harmonics for Switch-Mode Power Supplies

**Muhammad Naveed Iqbal** [1,*], **Lauri Kütt** [1], **Bilal Asad** [1,2], **Toomas Vaimann** [1,3], **Anton Rassõlkin** [1,3] **and Galina L. Demidova** [3]

[1] Department of Power Engineering and Mechatronics, Tallinn University of Technology, Ehitajate tee 5, 19086 Tallinn, Estonia; lauri.kutt@taltech.ee (L.K.); bilal.asad@aalto.fi (B.A.); toomas.vaimann@taltech.ee (T.V.); anton.rassolkin@taltech.ee (A.R.)

[2] Department of Electrical Engineering and Automation, Aalto University, 02150 Espoo, Finland

[3] Faculty of Control Systems and Robotics, ITMO University, 197101 Saint Petersburg, Russia; demidova@itmo.ru

[*] Correspondence: miqbal@taltech.ee



**Featured Application: The time-dependent variations in current harmonic emission from switch mode power supplies are often not considered during power quality measurements. It can affect the results of the current harmonic estimation models as switch-mode power supplies constitute a significant portion of domestic, commercial, and industrial electrical loads.**

**Abstract:** This paper presents the time-dependent variance in the current harmonics emission by power supplies during power quality measurements. Power quality problems are becoming more significant with the adoption of power electronic-based circuits such as power supplies. The switch-mode power supplies are widespread as industrial, commercial, and domestic electrical loads. They draw non-sinusoidal current from the utility and inject current harmonics. Therefore, they are the reason for poor power quality and reduction in the power factor. The current harmonics emission from these power supplies depends on the circuit topology, operating conditions, and filter inside them. The harmonic emission estimations are critical for network operators; however, various uncertainties have made it a complicated task. The time-dependent stability affects the magnitude and phase angle of the harmonic current measurements and estimation of power quality indices. This paper investigates the variation in current harmonics emitted by the power supply during the initial unstable period under constant load and operating conditions.

**Keywords:** power quality; switched-mode power supplies; thermal stability; total harmonic distortion; current harmonics

## 1. Introduction

Power quality issue is becoming critical for the network operators and electrical equipment manufacturers in the last few years. The electronics-based switching devices are burgeoning and escalating the current harmonics emission level in the electric supply system. With the enhancement in power electronics, the efficiency of domestic, commercial, and industrial electronic equipment has been boosted with a substantial reduction in their size. Almost every modern electrical equipment, such as personal computers (PC), battery chargers, household appliances, large commercial and industrial electrical systems encompasses a converter based power supply. These power supplies incorporate rectifiers and nonlinear components, thus polluting the distribution system with current harmonics. Therefore, current harmonic estimation is critical for power quality assessment. Probabilistic models

for current harmonic estimation are based on the measurements of the electrical loads [1] and most of the non-linear loads can be modeled as different switch-mode power supplies [2]. However, the time-dependent stability of the current harmonics generated by the power supplies may alter the outcome of such estimations. This paper deals with the deviation in the current harmonics magnitude and phase angles of switch-mode power supplies over time under constant load and input voltage.

The direct current (DC) power supply takes input from the utility or local DC source such as a battery. It maintains an interminable voltage level within the designed current limits and can be regulated or unregulated. The regulated power supplies uphold the output voltage close to the desired nominal value for the variations in voltage, load current, and temperature [3]. The switch-mode power supplies (SMPS) are regulated power supplies and contain alternating current to direct current (AC-DC) converters. Because of the nonlinear components in the SMPS, they are the source of power quality problems and feed current harmonics in the network.

These current harmonics prompt voltage distortion and the power factor reduction in the network. The current harmonics and voltage distortion cause overloading of the transformers. Their lifespan is reduced, and reliability is compromised. Excessive harmonic emission overheats electrical appliances and causes additional noise. Moreover, protection equipment, such as relays and circuit breakers, can also malfunction. Apart from polluting the network, these harmonics also affect the performance of SMPS itself [4]. Therefore, it is essential to estimate the impact of current harmonics emission in the network added by nonlinear power supplies. However, various uncertainties are associated with the power system that may affect the power quality estimations. Supply voltage variation and load operating modes can alter the harmonic emission in the grid [5]. The components present inside various electrical appliances may provide a variation because of material properties [6]. The aging of the components also affects the output [7]. The variation in power system operating conditions also results in variation of the network impedance [8,9]. However, uncertainties in the measurement of power quality indices like thermal stability of the components and transients are mostly ignored in the harmonic estimation models available in the literature.

Since SMPS are widely used in industrial, commercial and domestic loads, their collective impact could be devastating. Household loads in a residential distribution grid can be categorized into the linear load and the electronic load. However, electronic appliances are responsible for the majority of the current harmonic emission. These devices can be modeled as SMPS loads. Household appliances are categorized based on the SMPS circuit topologies in [2]. Consequently, for an accurate power quality assessment, every uncertainty during measurements of SMPS should be taken into account.

Several studies are available related to harmonic emission associated with the SMPS load. The harmonic cancellation between multiple SMPS operated at the same time has been presented in [10]. A Monte Carlo simulation is used to aggregate numerous SMPS loads. The author concludes that harmonic cancellation is more prominent in high power SMPS. The losses due to harmonic current injected by SPMS in a commercial building are discussed in [11]. The harmonic losses generated by modern electronics in the commercial buildings are responsible for neutral conductor overloading, overheating of the cables, and power factor reduction. The author projected an 8 kW of additional losses due to harmonic loading caused by SMPS. It will increase the total building wiring losses by up to 250%. The wiring loss caused by personal computers without harmonic elimination is 2.4 times compared to resistive loads. The comparison of electromagnetic interference in the grid due to series and parallel configuration of SMPS load is presented in another study [12]. A computer simulation is used to estimate the noise effectuated by SMPS in [13]. Topologies of SMPS are simulated to find the dimension that can impede noise emission defined by the standards. A Simulink model is used to scrutinize the current harmonics generated by the SMPS in another study [14]. The nonlinear loads can be imitated as parallel harmonic current sources with magnitude and phase angles [15]. A total harmonic distortion between 150% to 200% for different 3-phase balanced and unbalanced schemes was reported. The harmonic current injection was influenced by the number of equipment, circuit topologies, and type of equipment. The imbalance of the phase lines also altered the harmonic

injection. The current harmonic generation due to SMPS inside computers and other electronic equipment and the harmonic mitigation effects in the large office building is presented in [16].

Although, the studies mentioned above have considered uncertainties like harmonic cancellation, none of them discuss the effect of time-dependent variations of the harmonic currents due to thermal stability. A recent study reported significant variation in current harmonics magnitude and phase angles of LED lamps when pure sinusoidal voltage is applied [17]. LED lamps also contain rectifier circuits and draw a current waveform similar to the SMPS. The study concludes that by using measurements performed during the period when LED lamps were unstable leads to a significant error in harmonic current estimation.

This paper presents the results of the harmonic current variation of different industrial switch mode power supplies under constant load and operating conditions. A comprehensive overview of SMPS is presented in Section 2. The details of the experimental setup are described in Section 3. Section 4 includes a detailed analysis of the experimental results and framework to estimate the stability time of the power supplies. A comparison of the stability time of power supplies is also described in this section. Conclusions are presented in Section 5.

## 2. SMPS Overview

The SMPS converts AC to DC power by using switching devices, inductors, and capacitors. They deliver stable power to many industrial, commercial, and domestic electrical systems proficiently, as they contain rectifier circuits, consequently drawing non-sinusoidal current from the grid. As a result, they have low power factor and power quality [18]. SMPS are common in computers and battery chargers of different home appliances. A typical computer SMPS operates at 220V and draws input current with a total current harmonic distortion (THD$_i$) of 80%, and the power factor around 0.6 [19]. The block diagram of an SMPS is shown in Figure 1. The electromagnetic interference (EMI) filter reduces the high-frequency noise, and the inrush current limiter protects the circuit from the initial current surge due to capacitor charging. Most high power SMPS also contains a power factor correction circuits (PFC) to improve voltage regulations. The rectifier converts the input AC power to DC using switching devices like a diode, Insulated gate bipolar transistor IGBT, or MOSFET. The DC to DC converter is used to convert the input DC from one voltage level to another.

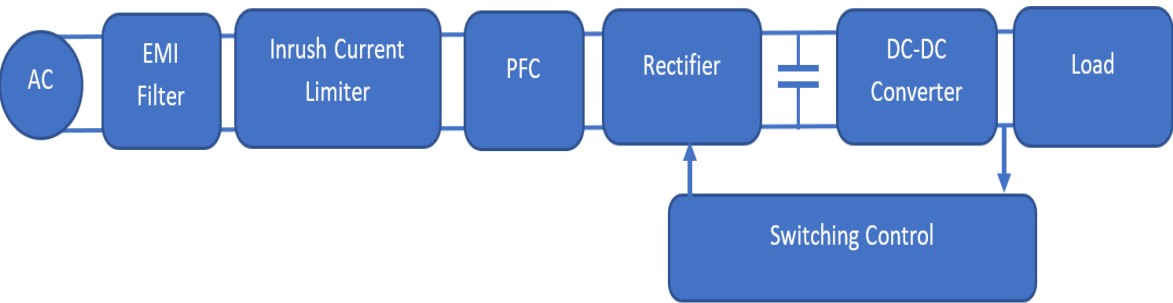

**Figure 1.** Block diagram of a typical switch-mode power supply.

The latest development in power electronics has enabled us to produce more efficient DC power supplies with improved power factor and stable output [20]. Modern design possesses low conduction losses and synthesizes near sinusoidal current waveform leading to fewer problems related to power quality. The SMPSs are classified based on the converters, power factor correction circuits, and type of switching control strategy [21]. The converter topology classification includes buck, boost, buck-boost, and multilevel converters [3]. The control strategies include pulse width modulation (PWM), proportional integral derivative (PID), sliding mode control (SMC), adaptive control, and neural network controllers. Many other control strategies are also employed and discussed in the literature [22]. The single-stage power supplies are widespread as they implement power conversion using a single switching circuit and simple control [23]. However, in computers and other

advanced applications, the single-stage power supplies are inadequate because of the stress across the switch and poor voltage regulations [22]. The SMPS in computers provides multiple DC outputs, and PFC circuits are used in the first stage for improving power factor and harmonics in the input current [24,25]. The impact of computer power supplies and other electronic equipment on the network has been discussed in [26]. Similar studies have also estimated the impact of computer power supplies and other new harmonic sources on the grid [27,28]. Most computer power supplies incorporate passive PFC circuits due to strict emission standards [29]. Researchers have proposed several designs with improved PFC for SMPS used in computer applications [30,31]

SMPS can be classified based on the type of filter circuits. Mostly low power SMPS (less than 75W) does not contain any PFC circuits. The high power SMPS may contain active or passive PFC. The devices with passive PFC contain a large inductor to smooth the variations in the current. On the other hand, devices with active PFC encompass an additional DC-DC converter. The SMPS with active PFC generates fewer harmonics than SMPS with passive PFC.

We have selected six single-phase SMPS from different manufacturers in the range of 30 W to 120 W. Table 1 shows the detailed specification of all six SMPS. The power supplies are categorized into two types based on the shape of the input current waveforms. Figure 2 shows the current waveform of both types of SMPS. Type 1 power supplies include passive filtering circuits to improve the power factor. The current waveform drawn by type 2 power supplies shows that they contain typical rectifier circuits without filters. We have applied 60% of the rated load current using the programmable electronic load to the power supply and recorded the input current drawn by the SMPS.

**Table 1.** Switch-mode power supplies (SMPS) characteristics.

| No. | Manufacturer | Model | Input Current AC (A) | Output Current DC (A) | Output Voltage DC (V) |
|-----|--------------|-------|---------------------|----------------------|----------------------|
| 1 | ABB, Zürich, Switzerland | CP-E | 0.83 | 5 | 24 |
| 2 | Dran, Chinfa Electronics, Taiwan | 120-24x | 0.63 | 5 | 24 |
| 3 | Entrelec,Germany | Systron 2A | 0.30 | 2 | 24 |
| 4 | Omron, Kyoto, Japan | S8VS-03024 | 0.60 | 1.3 | 24 |
| 5 | Siemens logo, Munich, Germany | 6EP1331-1SH03 | 0.70 | 1.3 | 24 |
| 6 | Siemens, Munich, Germany | 6EP1332-1SH71 | 0.67 | 2.5 | 24 |

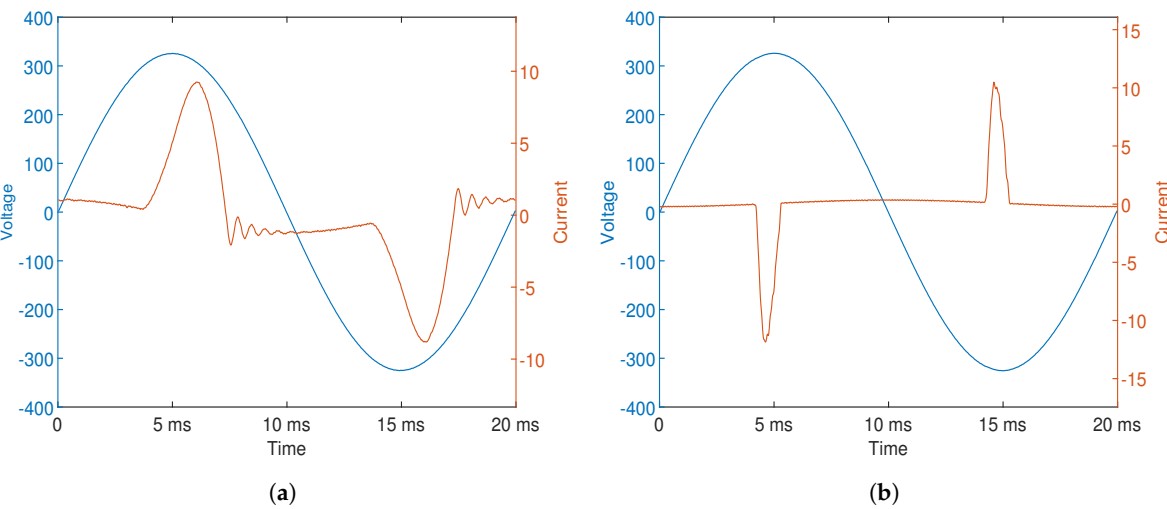

**Figure 2.** Current waveform drawn by SMPS (**a**) Type 1 (**b**) Type 2.

The current harmonic limits are defined in the International Electrotechnical Commission (IEC) 61000-3-2. The SMPS less than 600 W comes under class D equipment. Table 2 represents the maximum allowed harmonic current and harmonic currents per watt for odd harmonics [32].

**Table 2.** Current harmonic limits for class D devices.

| Harmonic Order (h) | Maximum Permissible Harmonic Current per Watt (mA/W) | Maximum Current Harmonic Limit (A) |
|---|---|---|
| 3 | 3.4 | 2.30 |
| 5 | 1.9 | 1.14 |
| 7 | 1 | 0.77 |
| 9 | 0.5 | 0.40 |
| 11 | 0.35 | 0.33 |
| $13 \leq h \leq 39$ | 3.85/h | 0.15(15/h) |

## 3. Measurement Setup

We have measured six SMPS for 60 min with our test bench and evaluated their $THD_i$ and current harmonics. The test bench involves a personal computer (PC), a 4kVA controllable power supply Chroma 61505 (Chroma system solutions, USA), a data acquisition (DAQ) module (National Instrumentation, United States), controllable electronic DC load TENMA 72-13210 (Premier Farnell, United Kingdom) and power quality measurement device a-eberle PQ-Box 200 (A-Eberle, Germany). The PQ-Box 200 can record at 200 ms resolution according to the standards IEC 61000-4-30 for CLASS A. These 200 ms data points are aggregated to 1-s interval data available to extract from the PQ-Box 200. Relays are used to switch all SMPS automatically. A control box is designed to provide 12V DC to the relays. The 50 Hz reference waveform for the power supply is generated with a sampling frequency of 100 kHz using MATLAB program through the DAQ module. The DAQ module is capable of generating an analog signal corresponding to the digital input. This reference signal ($V_{rf}$) has enabled us to generate a pure sinusoidal voltage through the programmable power supply as defined by Equation (1).

$$V_{rf} = \frac{V_o}{V_{range}} \times V_{co} \tag{1}$$

Here, $V_{range}$ is 300 V and $V_{co}$ is 7.072. The digital inputs to switch the relays are also generated using the same the MATLAB program via the DAQ module. The reference signal is calculated by using the relation indicated by Equation (2).

$$V_{out} = \sum_{i=1}^{n} \sqrt{2} \times A_i \sin(2\pi f_i t_s + \alpha_i) \tag{2}$$

$A_i$ and $\alpha_i$ are the root mean square values of the harmonics and its phase difference from the fundamental frequency, respectively. The harmonic frequency is shown by $f_i$ and sampling interval by $t_s$. The number of samples needed for the specific duration ($T_m$) of the voltage output from the controllable power supply can be calculated by Equation (3).

$$n = T_m \times f_s \tag{3}$$

Here $f_s$ is the sampling frequency and its value is 100 kHz. The controllable DC electronic load is used to operate each SMPS with 60% of its rated capacity. Figure 3 shows the block diagram of our test setup.

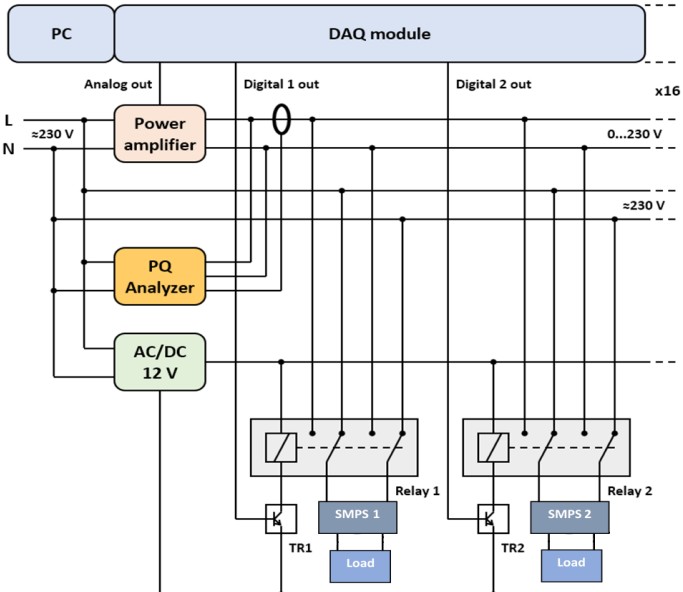

**Figure 3.** Block diagram of the measurement setup.

## 4. Results and Discussion

The measurement setup is used to test all six SMPS on pure sinusoidal voltage. The measurement is performed for 1 h, and 60% of the rated load is applied to each SMPS. The $THD_i$, current harmonics magnitude, and phase angles are analyzed. Figure 4 shows the absolute percentage difference between the $THD_i$ measured when the power supplies are in the cold state and stable state. The term "cold state" indicates the first five minutes of the measurements, and the "stable state" indicates the duration when the harmonic current variation is less than 0.25%. The first two power supplies indicate only 1.2% and 0.6% difference between the $THD_i$ measurements at the cold and stable state. However, the remaining power supplies have shown a significant difference between $THD_i$ values at the cold and the stable state. The fifth power supply shows the highest variance of 21%. Both third and fifth SMPS have a variation of 18 to 19% between cold and stable state. Similarly, the fourth SMPS shows a difference of 14.8%. Hence, Figure 4 indicates that type 1 SMPS shows a minimal difference in contrast to type 2 SMPS.

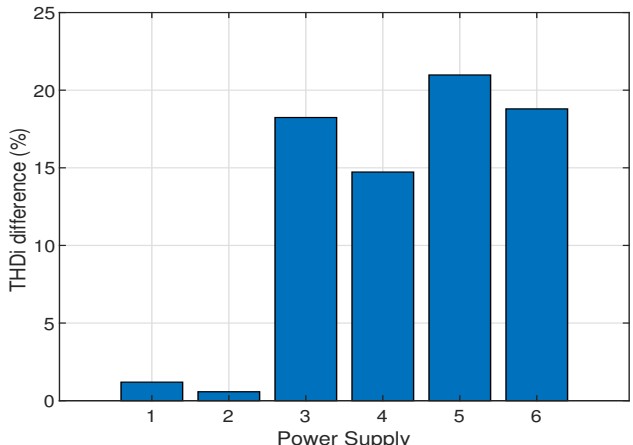

**Figure 4.** Histogram of total current harmonic distortion ($THD_i$) difference between cold and stable state of all power supplies.

To estimate the stability time of power supplies, we have used the curve-fitting approach to the current harmonic amplitude and phase angle variations over time. An exponential trend curve is

applied to the current harmonics magnitude, and phase angles up to the 19th harmonic. Equation (4) is used to calculate the parameters of the trend line using measurement data of the current harmonic magnitude and phase angle.

$$T_L = \triangle_i \times e^{\left(-\frac{i_t}{T_C}\right)} + \gamma_{stable} \tag{4}$$

$\gamma_{stable}$ is the current magnitude or phase value in the stable state. The delta $\triangle_i$ is the difference between the current harmonic magnitude or phase angle in the cold state and the stable state and calculated by Equation (5).

$$\triangle_i = \gamma_{stable} - \gamma_{cold} \tag{5}$$

The time constant in Equation (4) $T_C$ is the time taken by the exponential trend curve to increase or decrease by the factor e. It is calculated by using Equation (6).

$$T_C = \frac{T_{final} - T_{cold}}{1 - \ln(1 - \frac{1-(\gamma_{stable} - \gamma_{cold})}{\triangle_i}} \tag{6}$$

The stability time ($T_{stable}$) of the current harmonic magnitude and phase angle is three times of the time constant ($T_C$). Figure 5 shows the current magnitude variation of the fundamental and odd harmonics up to the 19th for the third power supply. The green line shows the measured harmonic current, and the blue line indicates the applied trend curve. The red dot indicates the time constant, and the black dot shows the stability time. A similar approach is applied to the phase angles of odd harmonics up to the 19th, as shown in Figure 6.

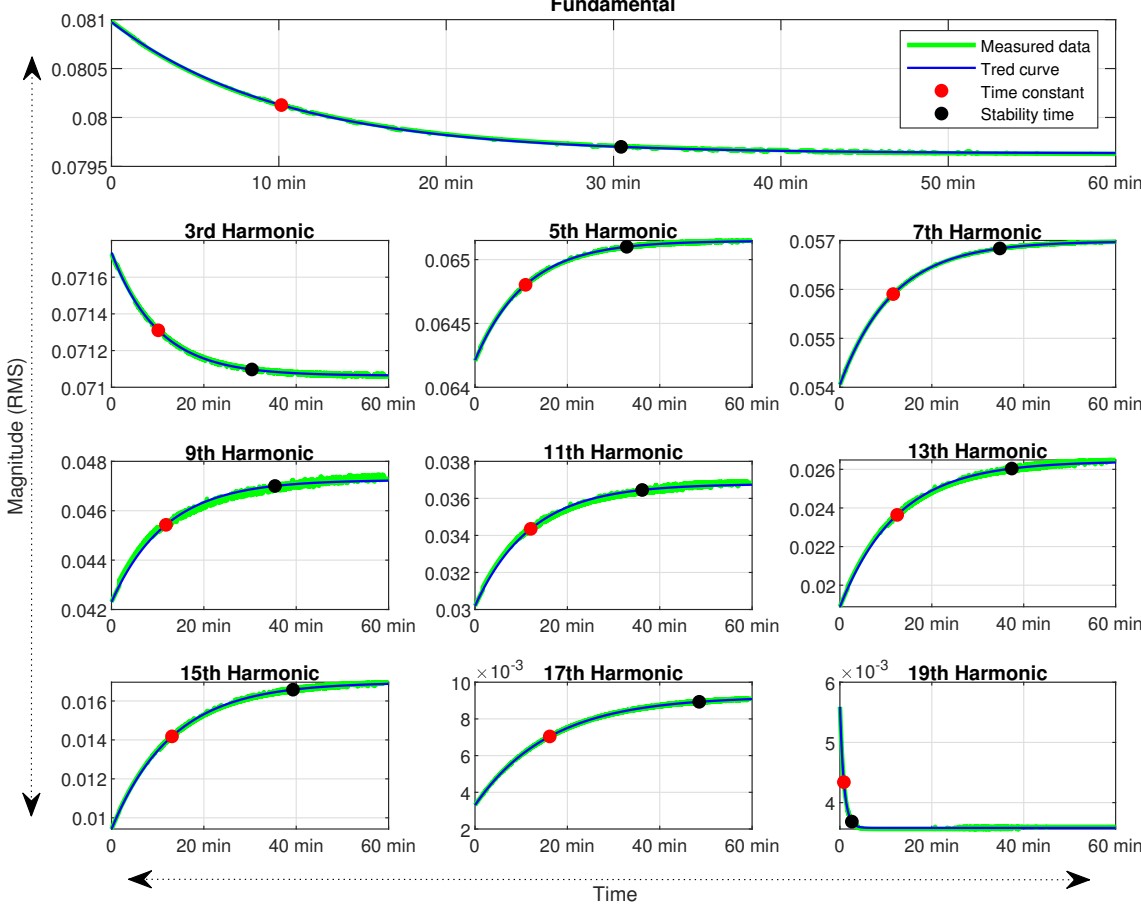

**Figure 5.** Current harmonic magnitude variation over time for power supply 3.

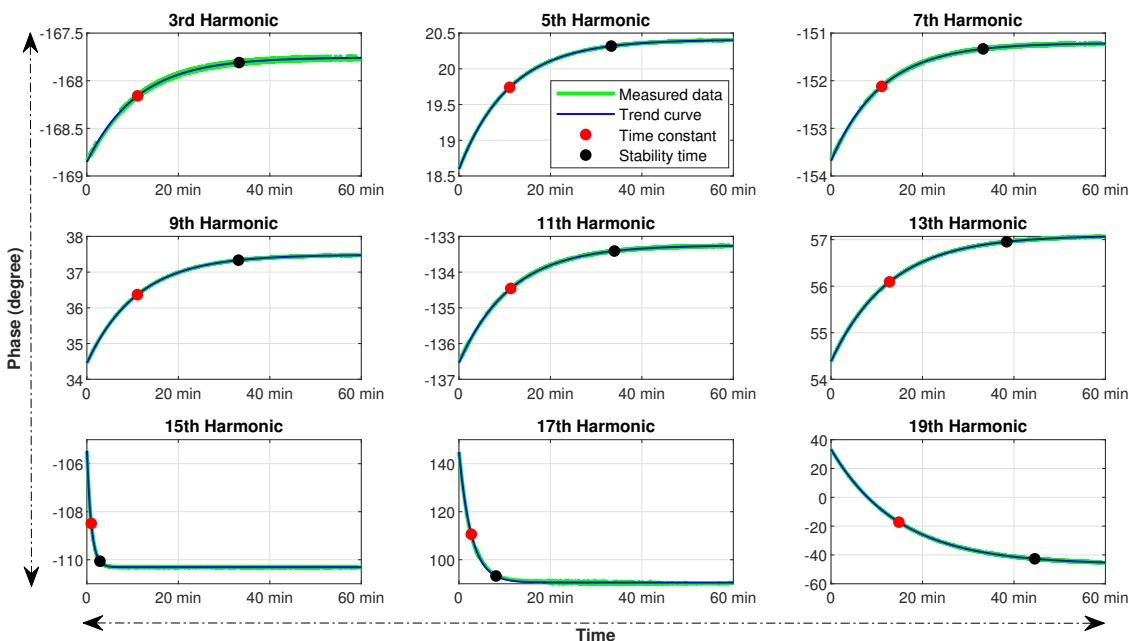

**Figure 6.** Current harmonic phase variation over time for power supply 3.

The difference between the cold and stable values of harmonic current magnitude and phase angles are more significant for type 2 power supplies. For type 1 power supplies, most of the current harmonics have less than 1% difference in the magnitude. The maximum difference observed for the fundamental and 7th harmonic for the first power supply is 1.2%. The power supply 2 shows the maximum difference of 1.7% for the 11th harmonic. Conversely, the type 2 power supplies exhibit a noteworthy difference, especially for the higher-order harmonics. The maximum difference was up to 194% for the 17th harmonic of power supply 3. Table 3 shows the minimum, maximum, and average deviation of cold and stable current harmonics magnitude for type 2 power supplies. The phase angle variation between cold and stable state is also less for type 1 power supplies. The first power supply only shows more than 1-degree variation for the 3rd, 5th, and 7th harmonic. On the other hand, power supply two shows phase angle variation of more than 1 degree for 15th, 17th and 19th harmonic. The type 2 power supplies show a significant deviation of phase angles between the cold and stable state measurements. Power supply three shows less phase angle variation of between 1–2 degrees among all type 2 power supplies. The other power supplies show a variation of 2 to 35 degrees, increasing from low to higher-order harmonics. Although the IEC 61000-3-2 standards do not define any limits for phase angle variation, the harmonic cancellation depends on current harmonics phase angles.

**Table 3.** Difference between cold and stable state current harmonics magnitude for type 2 SMPSs (%).

| Harmonics | Current Harmonics Magnitude Difference | | | | | | | | | |
|---|---|---|---|---|---|---|---|---|---|---|
| | **F** | **H3** | **H5** | **H7** | **H9** | **H11** | **H13** | **H15** | **H17** | **H19** |
| **Minimum** | 0.4 | 0.1 | 0.8 | 2.3 | 4.5 | 7.6 | 11.8 | 17.6 | 26.2 | 11.1 |
| **Maximum** | 1.6 | 1.4 | 3.7 | 7.5 | 13.4 | 23.8 | 43.4 | 88.2 | 194.7 | 97.0 |
| **Average** | 0.8 | 0.6 | 2.0 | 5.2 | 10.0 | 17.6 | 30.0 | 54.7 | 105.4 | 39.9 |

Figure 7a shows the magnitude stability time for all power supplies from fundamental to the 19th odd harmonic. Since the variation between cold and stable state magnitude values for type 1 power supplies is less than type 2, the stability time for type 1 power supplies is also less for most of the current harmonics. The power supply 1 and 2 have maximum stability time for the fundamental and

odd harmonics up to the 7th between 20 and 43 min. However, the 11th and 13th harmonic took more than 60 min to stabilize for the power supply 2. All other magnitudes of current harmonics for these two power supplies are less than the magnitude stability time of type 2 power supplies.

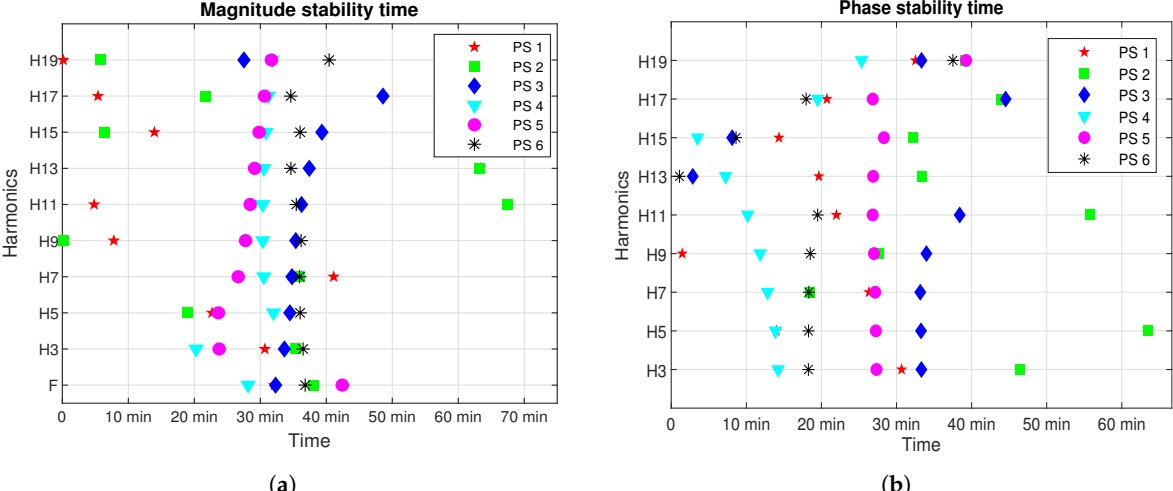

**Figure 7.** Stability time of current harmonics for all SMPSs (**a**) Magnitude (**b**) Phase.

The stability time for type 2 SMPS is in the range of 20 to 40 min, except for the 17th harmonic of power supply 3. The stability time of power supply 3 and 6 is quite close for most of the current harmonic magnitude. Similarly, the stability time of current harmonics magnitude for power supply 4 and 5 also follows the same trend but less than the power supply 3 and 6. Figure 7b shows the plot for phase angle stability for the odd harmonics for all switch-mode power supplies. Power supply 2 shows the highest value of stability time. For other power supplies, most of the current harmonics achieved stable phase angles before 35 min, except for the few high order harmonics.

The stability time is independent of the active power of the switch-mode power supplies. Figure 8 shows the graph of the active power of the power supplies against the stability time for all current harmonics magnitude up to the 19th harmonic. However, it is interesting to note that power supplies with high active power have less stability time on average in comparison to the power supplies with low active power consumption. This trend is more prominent for the higher-order harmonics.

The overall impact of thermal stability on harmonic current estimation could be significant in the real-time scenario. A similar study had indicated a notable difference for the harmonic current estimation for light-emitting diode (LED) lamps. The RMS current of LED lamps in a stable state was significantly different in comparison to the RMS current in a cold state [10].

Stochastic harmonic estimation models are based on the measurement results of the current harmonics of the electrical appliances. As thermal stability affects the measurement results, it may lead to a significant error in the harmonic analysis results. Figure 9 shows the percentage difference between the current RMS calculated at a cold and stable state for each power supply. Power supply 1 and 2 are type 1; therefore, it shows a small difference of about 1% between cold and stable states. All type 2 power supplies show a significant difference of more than 8% between current RMS values at a cold and stable state. Power supplies 5 and 6 show the maximal difference of more than 13%. Consequently, in a real-time scenario with many power supplies operating in an idle or working state, the estimation of current harmonics would result in an erroneous outcome because of their time-dependent variation.

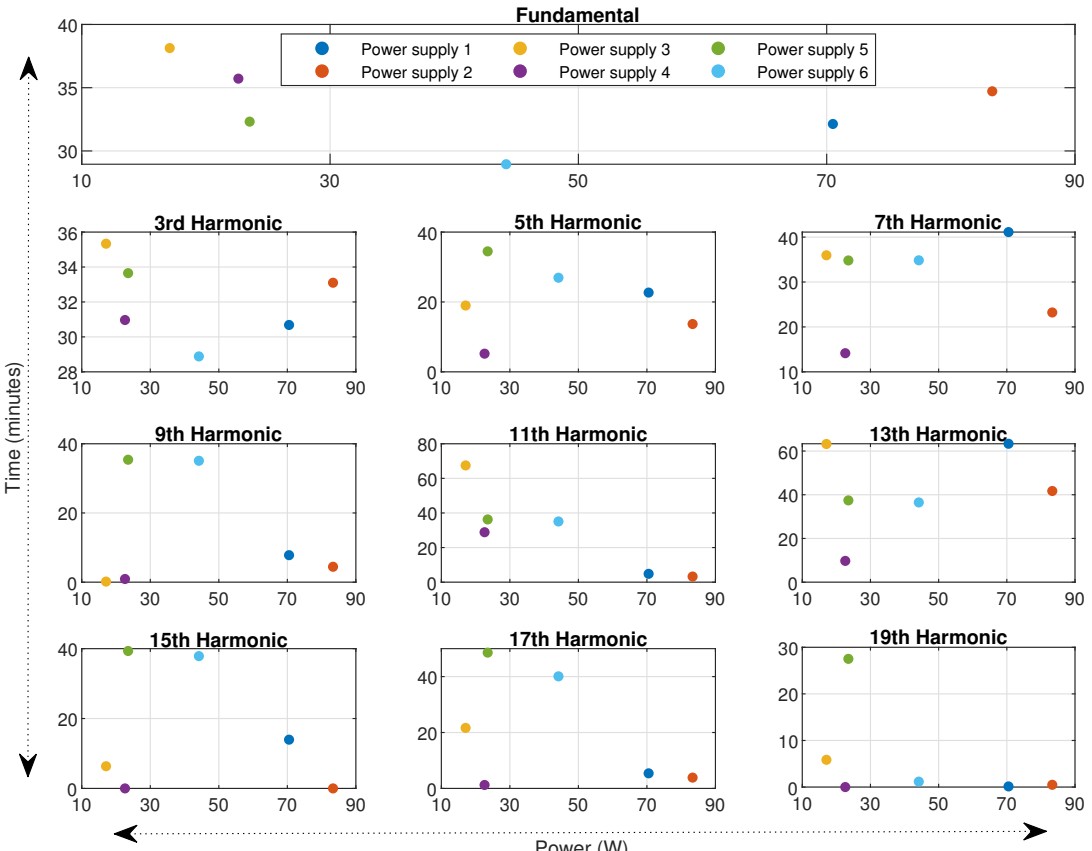

**Figure 8.** Current harmonic stability time ($T_{stable}$) variation against active power of all SMPS.

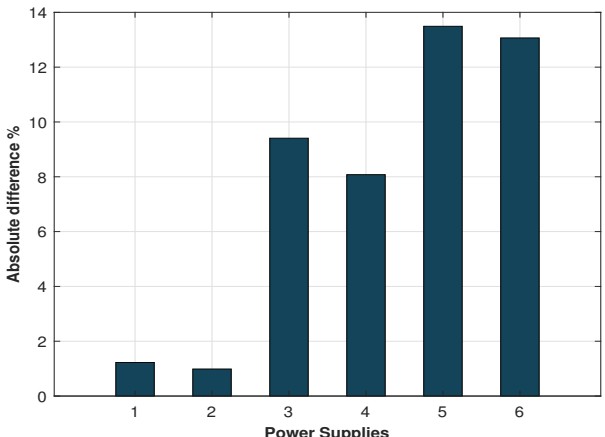

**Figure 9.** Current RMS difference between Cold and Stable state.

## 5. Conclusions

As the nonlinear loads are proliferating in the grid, the accurate assessment of power quality problems is becoming vital. The variation in current harmonics for the switch-mode power supplies has been discussed in this paper. The power supplies are categorized into two groups based on the filter circuits. The type 1 power supplies contain filtering circuits, while type 2 power supplies are without filters. The difference between cold and stable state measurements of current harmonics magnitude and phase angles are more significant for type 2 power supplies contrary to type 1. The THD$_i$ values of SMPS also vary with time until they become stable. The power supplies without filters show more difference between THD$_i$ measurements in the cold state and stable state. The THD$_i$ variation was between 15 to 20% for type 2 power supplies and 0.5 to 1.2% for type 1 power

supplies. The maximum time taken by power supplies to have stable current harmonics was 85 min. The absolute difference between cold and stable measurements was more prominent for higher-order harmonics for power supply without harmonic filters. For power quality estimation in the real-time scenario where a large number of power supplies have been involved, this difference may result in a significant error. Therefore, it is recommended to perform power quality measurements when the power supplies become stable. Future work will include power supplies with more rated power and other electronics-based equipment to assess time-dependent variation in harmonic emission.

**Author Contributions:** Conceptualization, M.N.I. and L.K.; methodology, M.N.I., B.A. and T.V.; validation, A.R. and M.N.I.; data curation, M.N.I. and G.L.D.; writing—original draft preparation, M.N.I. and B.A.; writing—review and editing, A.R. and G.L.D.; supervision, L.K. All authors have read and agreed to the published version of the manuscript.

**Funding:** This work was supported by the Estonian Research Council grant PSG 142. This work was financially supported by Government of Russian Federation, Grant 08-08.

**Conflicts of Interest:** The authors declare no conflict of interest.

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
