# Peer review of "Time Dependency of Current Harmonics for Switch-Mode Power Supplies"

_applsci, doi:10.3390/app10217806_

Round 1

Reviewer 1 Report

In general, I find the manuscript technically correct for publication, except for minor revision. I will only limit myself to giving the authors some recommendations for final review of document, which are explained next:

1- In Section 2 (SMPS overview), the authors should make a more complete explanation of block-diagram trying to focus the attention on the main functional blocks of SMPS that cause the greatest influence on power factor and harmonics of the input current (PFC, converter and switching control). Also, for SMPS computer applications, a greater number of specific works referenced in the literature should be sought. This way, the results of measurements obtained in Section 4 (Results and Discussion) would be more easily understood.

2- Several typographical errors have been made on the text of manuscript and/or Figures. I have detected the following:

-Line 72: <...define by the standards...> should be <...defined by the standards...>.

-Line 98: <...operates at 220 V draws input current...> should be <...operates at 220 V and draws input current...>.

 -Table 2: in the last row, "13≤h≥39" should be "13≤h≤ 39".

-Line 179: <...angels are more significant...> should be <...angles are more significant...>.

-Line 219: <...error in the harmoin analysis results...> should be <...error in the harmonic analysis results...>.

-Figure 9: label missing on vertical axis.

Author Response

Thank you for the valuable comments. The shortcomings are addressed and marked in blue color in the revised manuscript. 

1- The block diagram is explained. Detail about the computer SMPS and its impact on the power quality has been improved with literature references. 

2- The typographical errors mentioned are corrected and marked in blue color.

Reviewer 2 Report

This is an interesting paper. The experimental results provide interesting insight on the operation of non-linear harmonic load. The technical language of the paper can improve (e.g. non-linear current??)

Author Response

Thank you for your valuable comments. The technical language and errors are now corrected and marked in blue color in the revised manuscript.

Reviewer 3 Report

The article deals with the issues that we have been solving in my country since 1970. The occurrence of higher harmonics in phase-controlled thyristor sources (1970) led to complications with power factor compensation. The problem  is therefore old, but still unresolved. Therefore, I consider every contribution to this issue to be very valuable. Currently, the sources used are different, namely switch-mode power supply.

The article does not offer or describe a solution to the problem, it only presents the results of the harmonic current variation of a different industrial switch etc.

Unfortunately, only the power of 4 kVA controllable power was measured. It is true that the sources of these performances are the most widespread. Nevertheless, I believe that the general professional public should get acquainted with it.

Author Response

Thank you for the valuable comments and suggestions. 

I completely agree that the power quality issue has been in the highlights from 1970 and very valuable work has already been done. Many harmonic analysis models have been made and published by the researchers. However, the problem is still not addressed and as the amount of nonlinear load is increasing every day, this problem will become even worst in the near future. A lot of work is now being published related to the stochastic modeling of current harmonics. These models are based on certain assumptions and have limitations. In this paper, we have tried to highlight an issue related to the measurement of the harmonic currents. The thermal stability often ignored during the bottom-up harmonic current modeling and should be considered as it could lead to errors in the estimation of harmonic currents.